# The Computational Complexity of Counting Linear Regions in ReLU Neural Networks

**Moritz Stargalla**
University of Technology Nuremberg
`moritz.stargalla@utn.de`

**Christoph Hertrich**
University of Technology Nuremberg
`christoph.hertrich@utn.de`

**Daniel Reichman**
Worcester Polytechnic Institute
`daniel.reichman@gmail.com`

## Abstract

An established measure of the expressive power of a given ReLU neural network is the number of linear regions into which it partitions the input space. There exist many different, non-equivalent definitions of what a linear region actually is. We systematically assess which papers use which definitions and discuss how they relate to each other. We then analyze the computational complexity of counting the number of such regions for the various definitions. Generally, this turns out to be an intractable problem. We prove NP- and #P-hardness results already for networks with one hidden layer and strong hardness of approximation results for two or more hidden layers. Finally, on the algorithmic side, we demonstrate that counting linear regions can at least be achieved in polynomial *space* for some common definitions.

## 1 Introduction

Neural networks with rectified linear unit (ReLU) activations are among the most common and fundamental models in modern machine learning. The functions represented by ReLU networks are *continuous and piecewise linear* (CPWL), meaning that the input space can be partitioned into finitely many pieces on each of which the function is affine. Such pieces are called *linear regions*. This leads to the following intuition: the more linear regions a neural network can produce, the more complex problems it is capable of solving. Consequently, starting with Pascanu et al. [2014] and Montúfar et al. [2014], the number of linear regions became a standard measure of the expressive power of a ReLU network. Substantial effort has been put into understanding this quantity, e.g., by deriving upper and lower bounds depending on the network architecture or by developing algorithms to count it. More information can be found in the surveys Huchette et al. [2023], Balestriero et al. [2025].

Despite the significant interest in understanding the number of linear regions, surprisingly little is known about the most natural associated computational complexity question: Given a neural network, what are the time and space requirements needed to determine how many regions it has? The main objective of our paper is to make progress on this question by proving complexity-theoretical results on the problem of counting linear regions.

However, before one can even talk about counting linear regions, one has to properly define them. What sounds like a simple exercise is actually a non-trivial task. In the literature, there exists a variety of non-equivalent definitions of what counts as a linear region of a ReLU network. For example, some authors define it via possible sets of active neurons, others define it solely based on the function represented by the neural network. Some authors require regions to be full-dimensional, or connected,

39th Conference on Neural Information Processing Systems (NeurIPS 2025).

or even convex, others do not. Inconsistencies between definitions have led to confusion and even minor flaws in previous work, as we explain in Appendix A.4.

## 1.1 Our Contributions

**Definitions of linear regions**. In order to raise awareness to the technical, but important and non-trivial inconsistencies regarding the definition of linear regions in neural networks, we identify six non-equivalent, commonly used definitions in Section 3. We discuss how they relate to each other and provide a table demonstrating which authors used which definitions in previous work. We do not make a recommendation about what definition is the most reasonable one to use, as this depends on the context, but we encourage all authors of future papers to be aware of the subtleties carried by the different options and to be explicit about which definition they use and why.

**Complexity of counting regions in shallow networks**. As for many questions regarding ReLU networks, it makes sense to first understand the most basic case with one hidden layer. In Section 4, we prove that, regardless of which of the six definitions one uses, the seemingly simple question of deciding whether a shallow network has more than one linear region can indeed be decided in polynomial time. However, for all six definitions, we show that determining the exact number of regions is #P-hard, meaning that, unless the commonly believed conjecture #P $\neq$ FP fails, one cannot count regions of a shallow network in polynomial time. Furthermore, our reduction shows that even finding an algorithm that approximately counts the number of regions for one hidden layer might be intractable, as it would resolve long-standing open questions in the context of counting cells of hyperplane arrangements [Linial, 1986].

**Complexity of counting regions in networks with more than one hidden layer**. Wang [2022] showed that deciding if a deep neural network has more than $K$ regions is NP-hard. In Section 5 we improve upon Wang [2022] in several aspects. While the hardness by Wang [2022] only applies to networks with logarithmically growing depth (in the input dimension), we show that hardness can be proved for every constant number of hidden layers $\geq 2$ and even in the case $K = 1$, that is, for deciding if the network has more than one linear region. Our reduction also implies running-time lower bounds based on the exponential-time hypothesis. We furthermore show that, unless common complexity assumptions fail, one cannot even approximate the number of regions within an exponential factor in polynomial time.

**Counting regions using polynomial space**. While most of our results are concerned with lower bounds, in Section 6, we turn our attention towards proving an *upper* bound on the computational complexity of region counting. Wang [2022] proved[1] that for one definition of linear regions, the problem can be solved in exponential time. We show the stronger statement that for three of our definitions, polynomial space is sufficient.

**Limitations**. Our paper is of theoretical nature and we strive towards a thorough understanding of the problem of counting regions from a computational complexity perspective. As such, we naturally do not optimize our algorithms and reductions for efficiency or practical use, in contrast to, e.g., Serra et al. [2018] and Cai et al. [2023]. Our hardness results are of worst-case nature. Consequently, although beyond the scope of our paper, it is conceivable that additional assumptions render the problem tractable. For example, it would be very interesting to devise algorithms for region counting on networks that have been trained using gradient descent, as there is evidence that such networks have fewer regions [Hanin and Rolnick, 2019], which might allow faster algorithms. Not all of our results are valid for all of the six definitions we identify. We discuss the open problems resulting from this in the context of the respective sections. In our list of definitions in Section 3 and the corresponding Table 1, we tried to capture the most relevant previous works on linear regions, but a full literature review, like Huchette et al. [2023], is beyond the scope of our paper.

## 1.2 Related work

Huchette et al. [2023] survey polyhedral methods for deep learning, also treating the study of linear regions in detail. To the best of our knowledge, the first bounds on the number of regions in terms of the network architecture (e.g., number of neurons, network depth) were developed by Pascanu et al. [2014] and Montúfar et al. [2014]. Subsequently, better bounds were established [Raghu et al., 2017,

---

[1]The proof by Wang [2022] works for a different definition than claimed in their paper; see Appendix A.4.

Arora et al., 2018, Serra et al., 2018, Zanotti, 2025a]. Arora et al. [2018] prove that every CPWL function can be represented by a ReLU network.

Several works have developed algorithms for enumerating linear regions. Serra et al. [2018] and Cai et al. [2023] present mixed-integer programming based routines to count the number of regions and Masden [2025] presents an algorithm to enumerate the full combinatorial structure of activation regions. As discussed above, Wang [2022] provides some initial results on the computational complexity of counting regions, which we strengthen significantly in this paper. Our reductions are related to other decision problems on trained neural networks, e.g., verification [Katz et al., 2017], deciding injectivity or surjectivity [Froese et al., 2025a,b] or deciding whether the Lipschitz constant of a ReLU network exceeds a certain threshold [Virmaux and Scaman, 2018, Jordan and Dimakis, 2020].

Another line of research has studied the question of how to construct ReLU networks for functions with a certain number of regions [He et al., 2020, Chen et al., 2022, Hertrich et al., 2023, Brandenburg et al., 2025, Zanotti, 2025b]. The number of regions of maxout networks was studied by Montúfar et al. [2022]. Note that all our hardness results hold for maxout networks, too, as maxout is a generalization of ReLU. Goujon et al. [2024] present bounds for general piecewise linear activation functions. The average number of linear regions was studied, among others, by Hanin and Rolnick [2019], Tseran and Montúfar [2021]. Our work is inspired by the aim to better understand complexity-theoretic aspects of neural networks; another well-studied question in that regime is the complexity of training [Goel et al., 2021, Froese et al., 2022, Froese and Hertrich, 2023, Bertschinger et al., 2023].

## 2 Preliminaries

For $n \in \mathbb{N}$, we write $[n] := \{1, \dots, n\}$. For a set $P \subseteq \mathbb{R}^n$, we denote by $\overline{P}$, $P^\circ$, and $\partial P$ its closure, interior, and boundary, respectively. The ReLU function is the real function $x \mapsto \max(0, x)$. For any $n \in \mathbb{N}$, we denote by $\sigma : \mathbb{R}^n \to \mathbb{R}^n$ the function that computes the ReLU function in each component.

**Polyhedra, CPWL functions, and hyperplane arrangements**. A *polyhedron* $P$ is the intersection of finitely many closed halfspaces. A *polytope* is a bounded polyhedron. A *face* of $P$ is either the empty set or a set of the form $\arg\min\{c^\top x : x \in P\}$ for some $c \in \mathbb{R}^n$. A *polyhedral complex* $\mathcal{P}$ is a finite collection of polyhedra such that $\emptyset \in \mathcal{P}$, if $P \in \mathcal{P}$ then all faces of $P$ are in $\mathcal{P}$, and if $P, P' \in \mathcal{P}$, then $P \cap P'$ is a face of $P$ and $P'$. A function $f : \mathbb{R}^n \to \mathbb{R}$ is *continuous piecewise linear* (CPWL), if there exists a polyhedral complex $\mathcal{P}$ such that the restriction of $f$ to each full-dimensional polyhedron $P \in \mathcal{P}$ is an affine function. If this condition is satisfied, then $f$ and $\mathcal{P}$ are *compatible*. A *hyperplane arrangement* $\mathcal{H}$ is a collection of hyperplanes in $\mathbb{R}^n$. A *cell* of a hyperplane arrangement is an inclusion maximal connected subset of $\mathbb{R}^n \setminus (\bigcup_{H \in \mathcal{H}} H)$. A hyperplane arrangement naturally induces an associated polyhedral complex with the cells being the maximal polyhedra of the complex.

**ReLU networks**. A *ReLU neural network* $N$ with $d \geq 0$ hidden layers is defined by $d + 1$ affine transformations $T^{(i)} : \mathbb{R}^{n_{i-1}} \to \mathbb{R}^{n_i}$, $x \mapsto A^{(i)}x + b^{(i)}$ for $i \in [d+1]$. We assume that $n_0 = n$ and $n_{d+1} = 1$. The ReLU network $N$ *computes* the CWPL function $f_N : \mathbb{R}^n \to \mathbb{R}$ with

$$f_N = T^{(d+1)} \circ \sigma \circ \cdots \circ \sigma \circ T^{(1)}.$$

The matrices $A^{(i)} \in \mathbb{R}^{n_i \times n_{i-1}}$ are called the *weights* and the vectors $b^{(i)} \in \mathbb{R}^{n_i}$ are the *biases* of the $i$-th layer. We say the network has *depth* $d + 1$ and *size* $s(N) := \sum_{i=1}^{d} n_i$. Equivalently, ReLU networks can also be represented as layered, directed, acyclic graphs where each dimension of each layer is represented by one vertex, called a *neuron*. Each neuron computes an affine transformation of the outputs of its predecessors, applies the ReLU function, and outputs the result. We denote the CPWL function mapping the network input to the output of a neuron $v$ by $f_{N,v} : \mathbb{R}^n \to \mathbb{R}$. If the reference to the ReLU network $N$ is clear, we abbreviate $f_{N,v}$ by $f_v$.

**Activation patterns**. Given a ReLU network $N$, a vector $a \in \{0,1\}^{s(N)}$ is called an *activation pattern* of $N$ if there exists an input $x \in \mathbb{R}^n$ such that when $N$ receives $x$ as input, the $i$-th neuron in $N$ has positive output (is active) if $a_i = 1$ and 0 if $a_i = 0$. Given an activation pattern $a \in \{0,1\}^{s(N)}$, the network collapses to an affine function $f_N^a : \mathbb{R}^n \to \mathbb{R}$, and each neuron $i$ outputs an affine function $f_{N,i}^a : \mathbb{R}^n \to \mathbb{R}$ ($f_{N,i}^a$ is the zero function if $a_i = 0$). Again, if the reference to the ReLU network $N$ is clear, we abbreviate $f_{N,i}^a$ by $f_i^a$.

**Encoding size**. We use $\langle \cdot \rangle$ to denote the encoding size of numbers, matrices, or entire neural networks, where we assume that numbers are integers or rationals encoded in binary such that they take logarithmic space. More details can be found in Appendix A.1.

**Computational Complexity**. We give an informal overview over some notions of computational complexity and refer to [Arora and Barak, 2009] for further reading. A function $f : \{0,1\}^* \to \{0,1\}$ is in P if $f$ is computable in polynomial time by a deterministic Turing machine, in NP if it is computable in polynomial time by a non-deterministic Turing machine, and in RP if it is computable in polynomial time by a randomized Turing machine that never outputs false positives and accepts a correct input with probability at least $1/2$. Intuitively, P contains problems that can be efficiently solved while NP contains those whose solutions can be efficiently verified. It widely believed that $P \neq NP$ and $RP \neq NP$ hold. A function $f : \{0,1\}^* \to \mathbb{N}$ is in #P if there is a polynomial time non-deterministic Turing machine, which has exactly $f(x)$ accepting paths for any input $x \in \{0,1\}^*$ and in FPSPACE if $f$ is computable by a deterministic Turing machine that uses polynomial space. A problem is called *hard* for NP (analogously, for #P) if all other problems in this class can be reduced to it in polynomial time, and *complete* if it is both hard and contained in the class itself.

## 3  Definitions of linear regions

In this section we extract the six most commonly used definitions of linear regions from the literature and discuss their relations alongside with important properties and subtleties. Table 1 provides an overview of which previous papers use which definitions.

The set of inputs that have the same activation pattern induce a subset of $\mathbb{R}^n$ on which $f_N$ is affine.

**Definition 1** (Activation Region). *Given a network $N$ and an activation pattern $a \in \{0,1\}^{s(N)}$ with support $I \subseteq [s(N)]$, the set $S_{N,a} = \{x \in \mathbb{R}^n : f_i^a(x) > 0 \text{ for all } i \in I, \ f_i^a(x) \leq 0 \text{ for all } i \notin I\}$ is an activation region of $N$. If the reference to the ReLU network is clear, we abbreviate $S_{N,a}$ by $S_a$.*

Activation regions can be open, closed, neither open nor closed, and full- or low-dimensional, see Figure 1 for some examples. The (disjoint) union of all activation regions is exactly $\mathbb{R}^n$. In particular, the number of activation regions equals the number of activation patterns. It is important to note that the term *activation region* is used ambiguously. For example, Hanin and Rolnick [2019] use the term to refer to only *full-dimensional* activation regions.

**Definition 2** (Proper Activation Region). *Given a ReLU network $N$, a* proper activation region *of $N$ is a full-dimensional activation region of $N$.*

While the previous two definitions depend on the neural network *representation* itself, the following four definitions depend only on the CPWL *function* represented by the ReLU network and are independent from the concrete representation.

**Definition 3** (Convex Region). *Given a ReLU network $N$ and a polyhedral complex $\mathcal{P}$ that is compatible with $f_N$, a* convex region of $N$ given $\mathcal{P}$ *is a full-dimensional polyhedron $P \in \mathcal{P}$. The*

| Paper | Definitions | Paper | Definitions |
|---|---|---|---|
| Pascanu et al. [2014] | 4 | Montúfar et al. [2022] | 4 |
| Montúfar et al. [2014] | 5 | Wang [2022] | 1, 5 |
| Raghu et al. [2017] | 1, 5 (*) | Cai et al. [2023] | 2 |
| Arora et al. [2018] | 5 | Hertrich et al. [2023] | 6 |
| Serra et al. [2018] | 1,2 (*) | Huchette et al. [2023] | 2 (*) |
| Hanin and Rolnick [2019] | 2, 4 | Goujon et al. [2024] | 3, 6 |
| He et al. [2020] | 3, 6 | Brandenburg et al. [2025] | 3, 6 |
| Rolnick and Kording [2020] | 2, 5 (*) | Masden [2025] | 2 |
| Tseran and Montúfar [2021] | 1,5 (*) | Zanotti [2025a] | 4, 6 |
| Chen et al. [2022] | 3, 5, 6 | Zanotti [2025b] | 4 (*) |

Table 1: List of papers that use one or several definitions. Additional notes on the papers marked with an asterisk can be found in Appendix A.3. Lezeau et al. [2024] use another definition that lies between Definitions 4 and 5, see Appendix A.4.

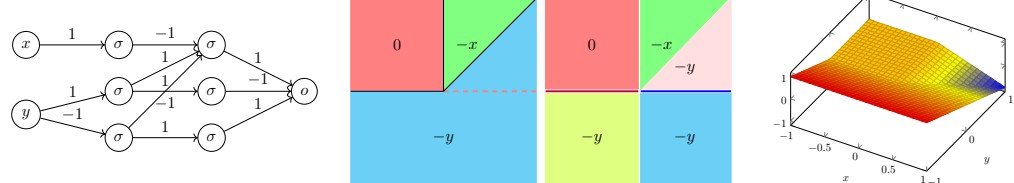

Figure 1: A ReLU network computing the function $f(x,y) = \max(-y, \min(0, -x))$. The closed connected regions (center left) and the activation regions (center right) are displayed. The slice $\{(x,0) : x \geq 0\}$ is contained in the two closed connected regions with functions $0$ (red) and $-y$ (blue). We have $R_6 = R_5 = R_4 = 3$, $R_3 = 4$, $R_2 = 5$ and $R_1 = 7$. In anti-clockwise direction starting from the region with value $0$, the activation patterns are $010110$, $000000$, $001001$, $101001$, $100000$, $110010$ and $110110$ (neurons are ordered from the upper left to the lower right).

number of convex regions *of $N$ is the minimum number of convex regions of any polyhedral complex $\mathcal{P}$ that is compatible with $f_N$.*

Note that many different polyhedral complexes can attain this minimal number. Hence, in general, it is not possible to refer to a polyhedron $P$ as a 'convex region of $N$' without specifying an associated polyhedral complex.

Another option to define linear regions is to use inclusion-maximal connected subsets on which the function computed by the ReLU network is affine, leading to the following definitions.

**Definition 4** (Open Connected Region). *Given a ReLU network $N$, an* open connected region *of $N$ is an open, inclusion-wise maximal connected subset of $\mathbb{R}^n$ on which $f_N$ is affine.*

**Definition 5** (Closed Connected Region). *Given a ReLU network $N$, a* closed connected region *of $N$ is a (closed) inclusion-wise maximal connected subset of $\mathbb{R}^n$ on which $f_N$ is affine.*

The subtle difference in the definition of open and closed connected regions has an important consequence: As Zanotti [2025a] showed, $\overline{P_1} \cap \overline{P_2} = \partial P_1 \cap \partial P_2$ holds for any distinct open connected regions $P_1, P_2$. Interestingly, the same is *not true* for closed connected regions. This is due to the fact that a closed connected region can continue on a low-dimensional slice of another closed connected region, which leads to a part of the boundary of one closed connected region to be contained in the interior of another closed connected region. Zanotti [2025a, Figure 1] gives a neat example where a low dimensional slice even connects two seemingly disconnected full-dimensional sets; another example can be found in Figure 1. Every open connected region is the interior of the closure of a union of some proper activation regions, see Lemma A.2. However, a closed connected region is in general not the closure of a union of some activation regions (see Appendix C.1).

Hanin and Rolnick [2019] define the set of open connected regions as the connected components of the input space where the set of points on which the gradient of $f_N$ is discontinuous are removed. Alternatively, the set of open connected regions is equal to the unique set $\mathcal{S}$ with the minimum number of open connected subsets such that $\bigcup_{S \in \mathcal{S}} \overline{S} = \mathbb{R}^n$ and $f_N$ restricted to any $S \in \mathcal{S}$ is affine, see Lemma A.1. The same is not true for closed connected regions, since there can be multiple sets $\mathcal{S}$ with the minimal number of closed connected subsets such that $\bigcup_{S \in \mathcal{S}} S = \mathbb{R}^n$ and $f_N$ restricted to any $S \in \mathcal{S}$ is affine. For example, in Figure 1, in such a minimal set $\mathcal{S}$ there is exactly one closed subset corresponding to the closed connected region with the constant zero function. There are multiple options to choose this subset, e.g. $(-\infty, 0] \times [0, \infty)$ or $((-\infty, 0] \times [0, \infty)) \cup \{(x, 0) : x > 0\}$.

By dropping the requirement of being connected, we obtain the following definition.

**Definition 6** (Affine Region). *Given a ReLU network $N$, an* affine region *of $N$ is an inclusion-wise maximal subset of $\mathbb{R}^n$ on which $f_N$ is affine.*

For each definition, a linear region $S \subseteq \mathbb{R}^n$ of a ReLU network $N$ can be associated with an affine function $g : \mathbb{R}^n \to \mathbb{R}$ such that $f_N(x) = g(x)$ for all $x \in S$. The affine function $g$ is unique if $S$ is full-dimensional. We say that the function $g$ is *computed* or *realized* on $S$. If $g$ is the zero function, we call $S$ a *zero region* and a *nonzero region* otherwise. The following theorem is immediate from the definitions.

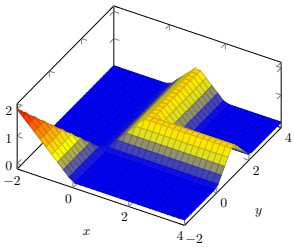 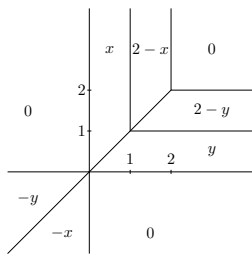

Figure 2: The function $\max(0, x) + \max(0, -y) - \max(0, x - y) + \min(\max(0, x - 2), \max(0, y - 2)) - 2\min(\max(0, x - 1), \max(0, y - 1))$. We have $R_6 = 7$, $R_5 = 8$, and $R_4 = 9$.

**Theorem 3.1.** *Given a ReLU network $N$, let $R_1, R_2, \ldots, R_6$ denote the number of activation regions, proper activation regions, convex regions, open connected regions, closed connected regions and affine regions, respectively. Then:* $R_6 \leq R_5 \leq R_4 \leq R_3 \leq R_2 \leq R_1$.

The examples in Figures 1 and 2 show that each inequality in Theorem 3.1 can be strict. Zanotti [2025a] showed $R_4 \in O((R_6)^{n+1})$. He et al. [2020] showed $R_3 \leq (R_6)!$. Trivially, $R_1 \leq 2^{s(N)}$.

**Problem definitions for counting linear regions**. In the remainder of the paper, we consider algorithmic problems arising from counting linear regions. Both decision problems (e.g., deciding if the number of regions is larger than a given threshold) or function problems (such as computing exactly or approximately the number of linear regions) are detailed below.

$K$-REGION-DECISION
**Input:** A ReLU network $N$.
**Question:** Does $N$ have strictly more than $K$ linear regions (according to a specified definition)?

LINEAR REGION COUNTING
**Input:** A ReLU network $N$.
**Question:** What is the number of linear regions of $N$ (according to a specified definition)?

## 4 Counting regions: one hidden layer

In this section, we derive our results for ReLU networks with one hidden layer. Our first main result is that 1-REGION-DECISION can be solved in polynomial time for ReLU networks with one hidden layer. Detailed proofs of the statements in this section are given in Appendix B.1.

**Theorem 4.1.** 1-REGION-DECISION *for networks with one hidden layer is in* P *for Definitions 1 to 6.*

The idea of the proof is as follows. In a ReLU network $N$ with one hidden layer, each neuron corresponds to a hyperplane that divides the input space into two halfspaces. It is not guaranteed that each hyperplane also leads to a discontinuity of the gradient of $f_N$, since the functions of the neurons with the same hyperplane may add up to an affine function. The proof of Theorem 4.1 shows that detecting whether a hyperplane of a neuron is canceled can be done in polynomial time. All hyperplanes of the network cancel if and only if the network computes an affine function and has thus only one linear region according to Definitions 3 to 6. For Definitions 1 and 2, 1-REGION-DECISION is trivial.

Froese et al. [2025b, Lemma 15] give a result similar to Theorem 4.1. They show that for a network with one hidden layer without biases, one can determine in polynomial time whether the network computes the constant zero function, and otherwise find a point on which the network computes a nonzero value. In contrast, Theorem 4.1 considers biases and nonzero affine functions.

Turning to the problem of exactly counting the number of regions, we show the following theorem.

**Theorem 4.2.** LINEAR REGION COUNTING *for ReLU networks with one hidden layer is* #P-*hard for Definitions 5 and 6 and* #P-*complete for Definitions 1 to 4.*

*Proof sketch.* Containment in #P is easy for Definitions 1 and 2, since an activation pattern $a \in \{0, 1\}^{s(N)}$ of a ReLU network $N$ is a unique certificate for a (proper) activation region, which can

be verified in polynomial time by computing the dimension of the set $S_a$, see Lemma A.5. More modifications are necessary to show #P containment also for Definitions 3 and 4.

To prove #P-hardness, we reduce from the problem of counting the number of cells of a hyperplane arrangement which is #P-complete, see [Linial, 1986]. Starting from a hyperplane arrangement $\mathcal{H}$ in $\mathbb{R}^n$, we carefully construct a neural network whose linear regions exactly correspond to the cells of the hyperplane arrangement. With proper technical adjustments, this works for all six definitions. □

Linial [1986] proved the #P-completeness of counting the number of cells of a hyperplane arrangement by reducing from the #P-complete problem of counting the number of acyclic orientations of a graph. His reduction implies that LINEAR REGION COUNTING remains #P-hard even for networks with one hidden layer where $A^{(2)} = (1, \ldots, 1)$, $b^{(1)} = 0$, $b^{(2)} = 0$, and $A^{(1)}$ is the transpose of an incidence matrix of a directed graph.

It is an open problem whether LINEAR REGION COUNTING is in #P for Definitions 5 and 6. Notice that a single activation pattern does not suffice as a certificate, since two proper activation regions with a non-empty intersection can have the same affine function. For example, consider the function $\max(0, x) + \max(0, -y) - \max(0, x - y)$ with zero regions $(-\infty, 0] \times [0, \infty)$ and $[0, \infty) \times (-\infty, 0]$.

To the best of our knowledge, it is unknown whether there is a polynomial factor approximation algorithm for approximating the number of cells in a hyperplane arrangement. Thus, it is also an open problem whether LINEAR REGION COUNTING has a polynomial factor approximation algorithm that runs in polynomial time.

## 5 Counting regions: going beyond one hidden layer

Here, we prove hardness results for ReLU networks with more than one hidden layer. Detailed proofs of the statements in this section are given in Appendix B.2, together with more detailed discussions providing additional intuition for some of the proofs.

### 5.1 Hardness of the decision version

From a result of Wang [2022], the following theorem follows immediately.

**Theorem 5.1** ([Wang, 2022]). *For any fixed constant $K \in \mathbb{N}_{\geq 1}$, $K$-REGION-DECISION for ReLU networks of depth $\Theta(\log n)$ is NP-hard according to Definitions 3 to 6.*

In their reduction from 3-SAT, they construct a network computing a minimum of $n + 1$ terms. As known constructions for computing the minimum require depth $\Theta(\log n)$, this leads to hardness for counting regions of networks with depth $\Theta(\log n)$. With Theorem 5.2, we improve on the result by showing that the problem remains NP-hard even for networks with two hidden layers.

**Theorem 5.2.** *For any fixed constants $K, L \in \mathbb{N}_{\geq 1}$, $L \geq 2$, $K$-REGION-DECISION for ReLU networks with $L$ hidden layers is NP-hard for Definitions 3 to 6.*

As a consequence, we even obtain hardness of the question whether there exists more than a single region. Proving this special case is also the first step of proving Theorem 5.2, as captured by the following lemma for the special case $K = 1$ and $L = 2$.

**Lemma 5.3.** *1-REGION-DECISION for ReLU networks with two hidden layers is NP-complete according to Definitions 3 to 6.*

*Proof sketch.* We reduce from SAT. Given a SAT formula $\phi$, we carefully construct a neural network $N_\phi$ with two hidden layers that has nonzero regions contained in $\varepsilon$-hypercubes around (0-1) points that satisfy $\phi$ and is constantly zero anywhere else. In this way, if $\phi$ is unsatisfiable, then $N_\phi$ computes the constant zero function and has exactly one linear region. If $\phi$ is satisfiable, then $N_\phi$ has strictly more than one linear region (one zero region and at least one nonzero region). □

Given a 3-SAT formula $\phi$ with $m$ clauses, the network $N_\phi$ from the reduction in the proof of Lemma 5.3 has input dimension and width $\mathcal{O}(m)$, whereas the network that is created in the reduction of Theorem 5.2 has input dimension $\mathcal{O}(m)$ and width $\mathcal{O}(m + K)$. We note that there is an alternative way to prove the NP-hardness of Lemma 5.3. Froese et al. [2025b, Theorem 18] show that the

problem of deciding whether or not a network without biases with one hidden layer has a point which evaluates to a positive value is NP-complete. By taking the maximum of the output of the network used in their reduction with the zero function, we obtain the NP-hardness of Lemma 5.3. However, our reduction offers a new perspective on the difficulty of the problem. In fact, the ideas used in our reduction are built upon in Section 5.2 to obtain results on the hardness of approximation of LINEAR REGION COUNTING. Moreover, our reduction has different properties, for example, all nonzero linear regions are bounded. This is not possible without biases, since then, all nonzero regions correspond to a union of polyhedral cones.

Theorem 5.2 can be proven using Lemma 5.3 in two steps. First, we can extend the hardness result of Lemma 5.3 from $1$-REGION-DECISION to $K$-REGION-DECISION by adding a new function with $K$ linear regions, and second, we can increase the number of hidden layers of the resulting network from 2 to $L$ by adding $L - 2$ additional hidden layers that compute the identity function.

As a corollary of Theorem 5.2, we obtain insights on the following decision problem.

$L$-NETWORK-EQUIVALENCE
**Input:** Two ReLU networks $N, N'$ with $L$ hidden layers.
**Question:** Do the networks $N$ and $N'$ compute the same function?

Two ReLU networks compute the same function if and only if the difference of the networks is the zero function. Since this difference can be computed by a single ReLU network, we obtain the following.

**Corollary 5.4.** $1$-NETWORK-EQUIVALENCE *is in* P, *and, for any fixed constant $L \geq 2$, $L$-NETWORK-EQUIVALENCE is* coNP-*complete*.

We also obtain the following runtime lower bound based on the Exponential Time Hypothesis.[2]

**Corollary 5.5.** *For any fixed constants $K, L \in \mathbb{N}, L \geq 2$, $K$-REGION-DECISION and* LINEAR REGION COUNTING *for Definitions 3 to 6 for ReLU networks with input dimension $n$ and $L$ hidden layers cannot be solved in $2^{o(n)}$ or $2^{o(\sqrt{\langle N \rangle})}$ time unless the Exponential Time Hypothesis fails.*

The $2^{o(n)}$ lower bound can be seen as another example of the curse of dimensionality in machine learning. As the input dimension grows, the problem quickly becomes intractable.

## 5.2 Hardness of exact and approximate counting

Here, we show that even *approximating* the number of linear region is hard for certain definitions. We prove two inapproximability results for different network architectures. For the first result, we use the proof ideas of Lemma 5.3 to show the following lemma.

**Lemma 5.6.** *For any fixed constant $L \in \mathbb{N}, L \geq 3$, there is a reduction from* #SAT *to* LINEAR REGION COUNTING *for networks with $L$ hidden layers according to Definitions 4 and 5.*

*Proof sketch.* Given a SAT formula $\phi$, the network $N_\phi$ from the proof of Lemma 5.3 has some nonzero linear regions contained in the $\varepsilon$-hypercube around every satisfying (0-1) point of $\phi$. In order to get control over the number of linear regions created per satisfying point, we carefully need to modify the network $N_\phi$ such that every satisfying assignment of $\phi$ creates the same number of nonzero linear regions. This yields a simple formula relating the number of linear regions of the ReLU network with the number of satisfying assignments of $\phi$. We achieve this by taking the minimum of the modified network with an appropriate function, which increases the number of hidden layers by one but does not change the width compared to $N_\phi$. The reduction can be extended to ReLU networks with $L \geq 3$ hidden layers as before. $\qquad\square$

We note that Lemma 5.6 does not hold for Definition 6, since the constructed network has multiple closed connected regions with the same affine function. Lemma 5.6 shows that a network having few regions does not necessarily imply that the regions of the network are "easy to count". On the contrary, it shows that instances with relatively few regions can lead to #P-hard counting problems.

---

[2] The Exponential Time Hypothesis [Impagliazzo and Paturi, 2001] states that 3-SAT on $n$ variables cannot be solved in $2^{o(n)}$ time.

The reduction from #SAT implies that *approximating* LINEAR REGION COUNTING is intractable as well, in the following sense: We define an *approximation algorithm* achieving approximation ratio $\rho \leq 1$ as an algorithm that is guaranteed[3] to return, given a network $N$ as input, a number that is at least $\rho$ times the number of regions of $N$. In fact, even though Lemma 5.6 only holds for Definitions 4 and 5, it is sufficient to prove the inapproximability result also for Definitions 3 and 6.

**Theorem 5.7.** *For any fixed constant $L \in \mathbb{N}, L \geq 3$, it is NP-hard to approximate LINEAR REGION COUNTING for Definitions 3 to 6 within an approximation ratio larger than $(2^n + 1)^{-1}$ for networks with $L$ hidden layers and input dimension $n$.*

*Proof.* If a SAT formula has no satisfying assignment, the network produced by the reduction of Lemma 5.6 will have exactly 1 linear region according to Definitions 3 to 6. Otherwise, it will have at least $1 + 2^n$ linear regions according to Definitions 3 to 6. If it was possible to achieve an approximation ratio larger than $(2^n + 1)^{-1}$ in polynomial time we could decide if a SAT formula is satisfiable in polynomial time. This concludes the proof. $\square$

We note that very similar ideas also rule out a fully polynomial randomized approximation scheme (FPRAS) for approximating the number of linear regions. A FPRAS [Jerrum, 2003] is a randomized polynomial-time (in the size of the input and $1/\varepsilon$) algorithm that returns a number $T$ such that $\text{Prob}[(1 - \epsilon)R \leq T \leq (1 + \epsilon)R] \geq 3/4$, where $R$ is the number of regions of the network. Theorem 5.7 can be easily adapted to show no FPRAS can exist for estimating the number of regions of neural network unless NP = RP.

Theorem 5.7 does not imply hardness of approximation for ReLU networks with two hidden layers. In the following, we show that approximation is indeed hard for networks with two hidden layers for Definitions 4 to 6, although with a weaker inapproximability factor than in Theorem 5.7.

**Theorem 5.8.** *Given a ReLU network with two hidden layers and input dimension $n$, for every $\varepsilon \in (0, 1)$ it is NP-hard to approximate LINEAR REGION COUNTING by a ratio larger than $2^{-O(n^{1-\varepsilon})}$ for Definitions 3 to 6.*

*Proof Sketch.* Given a SAT formula $\phi$, the network $N_\phi$ in the proof of Lemma 5.3 has exactly one linear region if $\phi$ is unsatisfiable and at least 2 linear regions if $\phi$ is satisfiable. We proceed by showing that for a ReLU network $N$ with $R$ linear regions, the network computing the function $f^{(k)} : \mathbb{R}^{nk} \to \mathbb{R}$, $f^{(k)}(x_{11}, ..., x_{1n}, ..., x_{k1}, ..., x_{kn}) = \sum_{i=1}^{k} f_N(x_{i1}, ..., x_{in})$ has exactly $R^k$ linear regions. In words, $f^{(k)}$ is simply the sum of $k$ copies of $f_N$ each having as input a disjoint set of $n$ variables. Applying this construction to $N_\phi$ for appropriate values of $k$ gives the desired result. $\square$

# 6 Counting regions using polynomial space

Due to the #P-hardness of LINEAR REGION COUNTING, we do not expect that efficient (polynomial time) algorithms for counting the number of linear regions exist. Wang [2022] claimed that LINEAR REGION COUNTING for Definition 5 would be in EXPTIME. As stated in Appendix A.4, the algorithm actually works for Definition 4 instead of Definition 5. Since closed connected regions are generally not a union of activation regions, it is still an open problem whether even an EXPTIME algorithm is possible for Definition 5. Also for Definition 3, to the best of our knowledge, it is not clear whether an EXPTIME algorithm exists, because there are infinitely many options to choose the underlying polyhedral complex. On the contrary, we show in this section that for Definitions 1, 2 and 6, the number of regions can be computed in polynomial space and therefore also in EXPTIME.

**Theorem 6.1.** LINEAR REGION COUNTING *is in* FPSPACE *for Definitions 1, 2 and 6.*

It is not hard to see that computing the number of activation regions and proper activation regions is possible in space that is polynomial in $\langle N \rangle$. Consider the following (informal) algorithm: Given a ReLU network $N$, iterate over all $2^{s(N)}$ vectors in $\{0, 1\}^{s(N)}$, and for each vector $a \in \{0, 1\}^{s(N)}$, compute the dimension of $S_a$ in time polynomial in $\langle N \rangle$ (see Lemma A.5) to determine if $S_a$ is an activation region or a proper activation region, and increase a counter by one if this is the case.

---

[3]The algorithm may be probabilistic and return the correct answer with probability bounded away from $1/2$.

Counting the number of affine regions is slightly more complicated, because a naive approach enumerating all the activation patterns would need to keep track of all the affine coefficients already seen to avoid double-counting, which is infeasible in polynomial space. Instead, we iterate over all possible affine functions by enumerating all possible coefficient combinations that have an encoding size of less than a polynomial upper bound, and check if there is a proper activation region on which the affine function is realized. The running time of this algorithm is exponential in the encoding size of the network, but it suffices to prove FPSPACE containment.

We describe how to count regions in Algorithm 2. The comments in the algorithms refer to the lemmas that show that the computation in the respective line can be performed in polynomial space.

---

**Algorithm 1** SEARCHAFFINEPIECE

---
**Input:** A ReLU network $N$ and a vector $(a_1, \ldots, a_n, b) \in \mathbb{Q}^{n+1}$.
**Output:** 1 if $\sum_{i=1}^{n} a_i x_i + b$ is a function of an affine region of $N$, else 0.
1: **for** $a \in \{0, 1\}^{s(N)}$ **do**
2:    **if** $\dim S_a = n$ **then**                                              ▷ (Lemma A.5)
3:       **if** $\sum_{i=1}^{n} a_i x_i + b = f_N^a(x)$ **then return** 1             ▷ (Lemma A.4)
   **return** 0

---

**Algorithm 2** EXHAUSTIVESEARCH

---
**Input:** A ReLU network $N$.
**Output:** Number of affine regions of $N$.
1: $n_{\max} = \max\{n_0, n_1, \ldots, n_{d+1}\}$
2: $U = 2^{36d^2 n_{\max}^2 \langle A_{\max} \rangle}$                                               ▷ (Lemma A.3)
3: $R = 0$
4: **for** $(a, b) \in \{-U, \ldots, U\}^{n+1} \times \{1, \ldots, U\}^{n+1}$ **do**
5:    **if** $\gcd(a_i, b_i) = 1$ for $i \in [n+1]$ **then**
6:       $R \leftarrow R + \text{SEARCHAFFINEPIECE}(N, (\frac{a_1}{b_1}, \ldots, \frac{a_{n+1}}{b_{n+1}}))$       ▷ (Lemma B.4)
   **return** $R$

---

The algorithms for counting (proper) activation regions (Definitions 1 and 2) show fixed-parameter tractability of LINEAR REGION COUNTING with respect to the number of neurons. Furthermore, recent results of Froese et al. [2025a, Corollary 4.4] imply W[1]-hardness of LINEAR REGION COUNTING for ReLU networks with two or more hidden layers when parameterized by the input dimension (for Definitions 3 to 6). However, the fixed-parameter tractability status of LINEAR REGION COUNTING remains open for other parameterizations and definitions.

## 7 Conclusion

We collected and discussed six commonly used non-equivalent definitions for linear regions of ReLU networks. We proved #P-hardness for counting the number of linear regions (for all six definitions) and NP-hardness for several associated decision problems (for most definitions). We further showed that for ReLU networks with two or more hidden layers, even approximating the number of linear regions is NP-hard (again, for most definitions). On the positive side, we showed that for some definitions, linear regions can at least be counted in polynomial space.

There remain many interesting open problems and directions for future work. Is LINEAR REGION COUNTING in EXPTIME for Definitions 3 and 5 (are there even finite algorithms)? Can some of the results in Sections 4 to 6 be extended to all six definitions? For example, is LINEAR REGION COUNTING for ReLU networks with one hidden layer contained in #P also for Definitions 5 and 6? Is the problem of approximating the number of linear regions also NP-hard for ReLU networks with one hidden layer? Finally, it would be interesting to study the fixed-parameter tractability of LINEAR REGION COUNTING under different parameterizations and definitions.

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
