# OpenReview forum: "The Computational Complexity of Counting Linear Regions in ReLU Neural Networks"
_NeurIPS.cc/2025/Conference — NeurIPS 2025 poster_

### Official Review · Reviewer_9Trg · 2025-06-08

**Clarity:** 2
**Significance:** 1
**Originality:** 3
**Rating:** 2
**Confidence:** 5

**Summary:**

This paper studies the computational complexity of counting linear regions of ReLU neural networks. It's shown that while such problem is computationally feasible for one-hidden-layer networks, it becomes inefficient for networks with more than one hidden layers.

**Questions:**

I believe this paper would be a better fit for TCS venues.

**Ethical Concerns:**

["NO or VERY MINOR ethics concerns only"]

**Final Justification:**

I have read through the author's response and other reviews. My main question on the relevance was not resolved by the author's reply. In the meantime, I agree with other reviewers that it's a technically solid paper, and there is certainly no harm to have it.

**Limitations:**

Yes.

**Quality:**

3

**Strengths And Weaknesses:**

**Main strength:** the results are novel and technically interesting. In particular, the authors show a sharp phase transition between one and more layers.

**Main weakness:** my main concern is on the relevance and significance to the machine learning community. (1) counting the number of linear regions is not a popular (and not a good one in my opinion, e.g. the highly self-repeating "saw-tooth" function) metric for the expressivity of neural networks. (2) deciding the computational complexity of such problem, is even less relevant, since I find no reason to consider it at all in practice.

**Other comments:** the improvement over the result of Wang 2022 seems incremental. The proof uses a similar framework, and the new regime covered isn't significant, since most practical neural networks have depth at least $O(\log n)$, already covered by Wang 2022. I believe it would be beneficial to discuss more on your technical novelties.

In section 3, you claim to unify the six definitions, while I found no statements supporting this claim (unless you can somehow show $R_1=O(R_6)$, or rewrite all six with a single notion). This makes this section look somewhat deviating.

---

> ### Author Rebuttal · Authors · 2025-07-30
>
> Thank you for carefully reading and evaluating our paper, and for your constructive feedback. We hope to have addressed all questions and comments in the responses below. If you have any further questions, please don’t hesitate to post them - we’ll be happy to clarify them.
>
> >my main concern is on the relevance and significance to the machine learning community. (1) counting the number of linear regions is not a popular (and not a good one in my opinion, e.g. the highly self-repeating "saw-tooth" function) metric for the expressivity of neural networks. (2) deciding the computational complexity of such problem, is even less relevant, since I find no reason to consider it at all in practice.
>
> There are several papers devoted entirely to studying and computing the number of linear regions of ReLU networks (e.g. Pascanu et al. [2014], Montufar et al. [2014], Serra et al. [2018], Hanin and Rolnick [2019], Wang [2022], Goujon et al. [2024]). In our view, this reflects a sustained interest in the topic within the machine learning community. More generally, the number of linear regions is one of the first and most fundamental measures to consider when trying to understand how complex a continuous piecewise linear (CPWL) function is. We do not believe that the relevance of this measure hinges only on Telgarsky’s “sawtooth” function. As evidenced by the cited works, a variety of algorithms have been proposed for counting linear regions. These algorithms for counting linear regions were, for example, used by researchers to investigate why certain architectural or training conditions (e.g. depth, number of training points) lead to a change in network performance. In this context, we see it as natural to study the computational complexity of counting linear regions.
>
> > the improvement over the result of Wang 2022 seems incremental. The proof uses a similar framework, and the new regime covered isn't significant, since most practical neural networks have depth at least $\mathcal{O}(\log n)$, already covered by Wang 2022. I believe it would be beneficial to discuss more on your technical novelties.
>
> While there are certainly architectures that are deeper, it seems likely that networks of lower depth could be used by commercial companies. In addition, researchers may want to estimate the number of linear regions (and have done so in the past) and use for this purpose wider networks of limited depth - for which our results apply, but those of Wang [2022] do not.
> We emphasize that our contributions are not incremental and outline our new technical novelties below:
> - In contrast to Wang [2022], we strengthen the hardness results from a depth that grows with input dimension to a constant depth. To achieve this, we build on the ideas from Katz et al. [2017] and construct a network that outputs 1 on a 0-1 input point if and only if the point represents a satisfying assignment to the given SAT instance, and 0 otherwise. The core technical novelty in this improvement lies in how we “nullify” this network’s output outside the vicinity of 0-1 points. While Wang [2022] achieves this by taking the minimum of $n+1$ functions using $\log(n+1)$ hidden layers to, we design a new “gadget function” that achieves this nullification using only a single hidden layer.
> - Beyond hardness of decision problems, we prove hardness of approximation, which was not addressed in prior work. In Lemma 5.6, we extend our reduction with an additional hidden layer and a second novel gadget function that ensures each satisfying 0-1 point leads to a fixed, predetermined number of linear regions.
> - In Theorem 5.8, we present a novel construction that amplifies inapproximability by summing multiple copies of a “hard” network.
> - Our results also cover six distinct definitions of linear regions, whereas prior work usually only considered one or two definitions. This generality required additional effort: for instance, Theorem 4.2 relies on a reduction from hyperplane arrangement cell counting, which requires carefully orienting hyperplanes to ensure the result holds for all definitions.
> - Finally, in Algorithm 2, we present a novel FPSPACE algorithm for counting affine regions. A key difficulty in this setting is avoiding the need of keeping track of all encountered affine functions explicitly, which would require exponential space. Our approach sidesteps this by enumerating all possible coefficient combinations, thereby iterating over all possible affine functions implicitly, using only polynomial space.
>
> >In section 3, you claim to unify the six definitions, while I found no statements supporting this claim (unless you can somehow show $R_1=\mathcal{O}(R_6)$, or rewrite all six with a single notion). This makes this section look somewhat deviating.
>
> The goal of Section 3 was not to assert that all definitions are equivalent or easily relatable to one another (beyond the bounds that we cited from prior work). Rather, the aim of the section is to systematically present the six most widely used definitions and to highlight the subtle but important distinctions between them. To avoid any possible confusion, we are happy to revise the opening sentence of Section 3 to more clearly reflect this intention. Note that it is not possible to obtain significantly better bounds than the trivial bound $R_1\in 2^{\mathcal{O}(s(N))} \cdot R_6$. To see this, consider a hyperplane arrangement $H$ in $\mathbb{R}^n$ with $n$ hyperplanes that are in general position. This hyperplane arrangement partitions the space into $2^n$ cells. We can construct a ReLU network $N$ with $2n$ ReLU neurons such that, for each hyperplane in the hyperplane arrangement, there are exactly two neurons inducing that hyperplane such that their contributions cancel out. The resulting network computes the zero function, and therefore has only one affine region (i.e., $R_6=1$). However, the network $N$ has at least $2^n$ activation regions ($R_1>=2^n$). Inserting $s(N)=2n$ yields $R_1>= 2^{s(N) / 2} \cdot R_6$.
>
> >I believe this paper would be a better fit for TCS venues.
>
> While the theoretical focus of the paper is indeed a limitation as we acknowledged, we do not view this as a weakness. In our view, pure theory papers like ours are a valuable contribution to NeurIPS. Similar computational complexity theory papers have been published at NeurIPS (e.g. Blum and Rivest [1988], Livni et al. [2014], Barceló et al. [2020], Bertschinger et al. [2023]).
>
> References.
> - Training a 3-node neural network is NP-complete, Blum and Rivest. NeurIPS 1988.
> - On the Computational Efficiency of Training Neural Networks. Livni et al. NeurIPS 2014.
> - Model Interpretability through the Lens of Computational Complexity. Barceló et al. NeurIPS 2020.

---

> > ### Comment · Reviewer_9Trg · 2025-08-05
> >
> > Thank you for the detailed response. I do agree with you that the notion of number of linear regions, although less popular, is a well-established one as a measure of expressivity of neural networks. However, the motivation of studying the computational complexity of deciding such metric, is not justified: typically the study of computational complexity is of interest when it's on a practical algorithmic objective. A well-established notion doesn't necessarily make the computational complexity of this notion well-established.
> >
> > I do appreciate your outline of technical novelties. Still, I'm not convinced that they are relevant to the ML community if the motivation of the problem considered is in question. Similarly, the existence of previous TCS papers at Neurips doesn't necessarily make s TCS-taste paper a good fit for the venue. For example, training neural networks is a much more practical algorithmic objective in my opinion, and studying its computational complexity is certainly relevant.

---

> > > ### Author Response · Authors · 2025-08-05
> > > **Response**
> > >
> > > Thank you for your comment. We would like to note that four papers we cite implement algorithms for finding the number of linear regions: Hanin and Rolnick, Wang, Serra et al., and Masden. Three of these papers have been published in ML venues. So we think there is interest in computing the number of linear regions in the ML community.

---

### Official Review · Reviewer_sbxs · 2025-06-27

**Clarity:** 4
**Significance:** 3
**Originality:** 3
**Rating:** 5
**Confidence:** 3

**Summary:**

This paper studies the computational complexity of counting the number of linear regions induced by ReLU neural networks. The authors identify and formalize six non-equivalent definitions of linear regions used in the literature. They prove that exact counting is #P-hard even for networks with one hidden layer, and that approximate counting is NP-hard for networks with two or more hidden layers. They also show that the decision problem (e.g., whether the number of regions exceeds a threshold) is NP-complete for networks with at least two hidden layers. Finally, they provide a positive result: exact counting is in FPSPACE for some of the definitions.

**Questions:**

1. Do the hardness results still hold in the absence of bias terms? It appears that the SAT-based reductions used in the paper rely on biases to construct bounded and localized nonzero regions. If the ReLU networks are restricted to be bias-free, do the same hardness results—particularly the NP-hardness and inapproximability for deep networks—still apply? Clarifying this would help understand whether the complexity arises inherently from the activation patterns or from representational power added by biases.
2. Have the authors considered the parameterized complexity of counting linear regions? While the paper establishes strong worst-case hardness results (e.g., #P-hardness for 1-layer networks and ETH-based lower bounds in input dimension), it would be interesting to analyze whether the problem admits fixed-parameter tractable algorithms with respect to natural parameters such as the total number of neurons s(N), the network depth d, or the number of regions R.

**Ethical Concerns:**

["NO or VERY MINOR ethics concerns only"]

**Final Justification:**

I appreciate the authors’ detailed rebuttal and clarifications. The paper makes a valuable contribution by formalizing multiple definitions of linear regions and establishing strong worst-case hardness results for counting them. The additional discussion on bias-free networks and parameterized complexity is helpful, though these directions remain largely open. My earlier concerns regarding the practical relevance of the results, given their focus on worst-case complexity, remain, but I agree that such results are an essential foundation before studying trained networks or special cases. Overall, the technical contributions are solid and well-presented, and I maintain my original score.

**Limitations:**

1. The paper exclusively analyzes the worst-case computational complexity of counting linear regions. While this is important from a theoretical standpoint, it may not fully capture the behavior of ReLU networks encountered in practice—particularly those trained using gradient-based methods, which are often observed to induce relatively few linear regions.

2. Although the authors show that region counting is in FPSPACE for some definitions, the algorithms presented are mainly of theoretical interest and not designed for practical implementation. There is little discussion of heuristics, approximations, or tractable special cases.

**Paper Formatting Concerns:**

No.

**Quality:**

3

**Strengths And Weaknesses:**

Strengths:
1. The paper offers a clear and systematic organization of existing literature by identifying and formalizing six non-equivalent definitions of linear regions, resolving ambiguities that have led to inconsistencies in prior work.
2. It establishes new and strong computational hardness results, including #P-hardness for shallow networks and NP-/inapproximability results for deep networks under various definitions.

Weaknesses:
1. The results are based entirely on worst-case complexity analysis, which may not fully reflect the structure of practically trained networks. For instance, gradient-based training often induces fewer regions, potentially making counting easier in practice.
2. The parameterized complexity of the problem is not explored; fixed-parameter tractability in terms of the number of neurons, depth, or region count remains an open direction.

---

> ### Author Rebuttal · Authors · 2025-07-30
>
> Thank you for carefully reading and evaluating our paper, and for your constructive feedback. We hope to have addressed all questions and comments in the responses below. If you have any further questions, please don’t hesitate to post them - we’ll be happy to clarify them.
>
> >1. The results are based entirely on worst-case complexity analysis, which may not fully reflect the structure of practically trained networks. For instance, gradient-based training often induces fewer regions, potentially making counting easier in practice.
> >2. The parameterized complexity of the problem is not explored; fixed-parameter tractability in terms of the number of neurons, depth, or region count remains an open direction.
>
> We agree that proving results for trained networks is very interesting. This itself presents a major challenge and we are not aware of previous hardness results for such trained networks. In our view, before trying to prove results for special cases it seems essential to first prove worst-case results. We will explicitly mention the parameterized complexity question in the camera-ready version of the paper. We also think that it is important to distinguish between worst-case instances in terms of computational complexity (“hard to count”) and those in terms of the number of linear regions (“many regions”). The relationship between these two notions is not straightforward - a network having fewer regions does not necessarily imply the regions of the network are “easy to count”. In fact, for Definitions 3 to 6, our reduction in Lemma 5.6 shows that instances with relatively few regions (compared to the worst-case number of linear regions) can lead to #P-hard counting problems.
>
> >1.Do the hardness results still hold in the absence of bias terms? It appears that the SAT-based reductions used in the paper rely on biases to construct bounded and localized nonzero regions. If the ReLU networks are restricted to be bias-free, do the same hardness results—particularly the NP-hardness and inapproximability for deep networks—still apply? Clarifying this would help understand whether the complexity arises inherently from the activation patterns or from representational power added by biases.
>
> It seems that our SAT-based reductions likely extend to the case without biases. Note that by replacing all biases $b$ in the network with $b \cdot |y|$, where $y$ is an extra input coordinate, we obtain a bias-free network (albeit with one extra hidden layer, as $|y|=\max(0,y)+\max(0,-y)$ needs to be computed at first). With this modification, the respective hardness results should hold also in the bias-free case: if the original network has $k$ affine regions, then the bias-free network has between $k$ and $2k$ affine regions, and the original network is equally zero if and only if the bias-free network is equally zero, but making this rigorous is subject to further research.
>
> >2. Have the authors considered the parameterized complexity of counting linear regions? While the paper establishes strong worst-case hardness results (e.g., #P-hardness for 1-layer networks and ETH-based lower bounds in input dimension), it would be interesting to analyze whether the problem admits fixed-parameter tractable algorithms with respect to natural parameters such as the total number of neurons s(N), the network depth d, or the number of regions R.
>
> While the (informal) algorithms presented in Section 6 for counting activation regions and proper activation regions (Definitions 1 and 2) show fixed parameter tractability of the corresponding problems with respect to the number of neurons $s(N)$, many important questions regarding the parameterized complexity of the problems remain open. As stated above, we will explicitly mention these questions in the camera-ready version of the paper.
>
> >1. The paper exclusively analyzes the worst-case computational complexity of counting linear regions. While this is important from a theoretical standpoint, it may not fully capture the behavior of ReLU networks encountered in practice—particularly those trained using gradient-based methods, which are often observed to induce relatively few linear regions.
>
> We have addressed this comment above.
>
> >2. Although the authors show that region counting is in FPSPACE for some definitions, the algorithms presented are mainly of theoretical interest and not designed for practical implementation. There is little discussion of heuristics, approximations, or tractable special cases.
>
> Theorems 5.7 and 5.8 show NP-hardness of approximation within a factor that is exponential in the input dimension.

---

> > ### Comment · Reviewer_sbxs · 2025-08-05
> >
> > Thanks for the reply. I will keep my score.

---

### Official Review · Reviewer_yJzi · 2025-06-29

**Clarity:** 3
**Significance:** 3
**Originality:** 3
**Rating:** 5
**Confidence:** 3

**Summary:**

The paper considers fully-connected ReLU neural networks. It is well-known that the function associated to such a network is piece-wise linear. The paper studies the number of pieces (called linear regions) in the input space, where the function is linear. The paper provides 6 precise mathematical definitions of linear regions from the literature. Then, for these definitions, the paper provides results of a Theoretical-Computer-Science (TCS) nature on the computational complexity of counting the linear regions. There are positive and negative results.

**Questions:**

Can you exploit the extra page, if there is a camera ready step for the paper, to add
- intuition about the Machine Learning impact of the results, especially on the differences between definitions
- pedagogical explanations on the complexity theory concepts, for a general audience

Why is it written "definitions X and Y" instead of "Definitions X and Y" in the paper?

**Ethical Concerns:**

["NO or VERY MINOR ethics concerns only"]

**Final Justification:**

Given the author's response to me, and to the other reviews, I am quite confident in my positive evaluation of the paper, in the end.

**Limitations:**

YES

**Paper Formatting Concerns:**

No concerns.

**Quality:**

3

**Strengths And Weaknesses:**

STRENGTHS
The paper appears to be mathematically precise and rigorous. I agree with the authors that being mathematically precise about the definition and manipulation of a linear region is important, as without clarity, mistakes are quickly made, as the authors point out in the literature review. The results obtained by the authors appear to be wide, as they cover 6 existing definitions of linear regions, as well as several notions of complexity from TCS. I think studying theoretical properties of ReLU networks is important since they are fundamental in deep learning.

WEAKNESSES The paper is a bit dry, probably because there are many results and because of the page limit. There does not seem to be machine learning intuition as to why different definitions lead to different computational complexities. The reading is not user-friendly to readers that are not experts in complexity theory, in my opinion.

---

> ### Author Rebuttal · Authors · 2025-07-30
>
> Thank you for carefully reading and evaluating our paper, and for your constructive feedback. We hope to have addressed all questions and comments in the responses below. If you have any further questions, please don’t hesitate to post them - we’ll be happy to clarify them.
>
> >The paper is a bit dry, probably because there are many results and because of the page limit.
>
> While we strongly believe that rigor is essential in theoretical work, we regret that the paper has come across as somewhat dry.
>
> >There does not seem to be machine learning intuition as to why different definitions lead to different computational complexities. The reading is not user-friendly to readers that are not experts in complexity theory, in my opinion.
>
> While some of our results do not rule out the possibility that the true hardness of the studied problems is the same for all definitions (e.g. Theorem 4.2), the hardness changes depending on the definitions for other problems. For example, the problem of deciding if a ReLU network with two hidden layers has two or more regions is NP-complete for Definitions 3 to 6 (Lemma 5.3) but is trivial for Definitions 1 and 2. The corresponding intuition is that (proper) activation regions (Def. 1 and 2) seem to have a simpler structure than the other definitions. We will add explanations on complexity theory concepts in the camera-ready version of the paper.
>
> >Can you exploit the extra page, if there is a camera ready step for the paper, to add
> >- intuition about the Machine Learning impact of the results, especially on the differences between definitions
> >- pedagogical explanations on the complexity theory concepts, for a general audience
>
> We will add additional intuition and explanations on complexity theory concepts for a general audience in the camera-ready version of the paper.
>
> >Why is it written "definitions X and Y" instead of "Definitions X and Y" in the paper?
>
> Thank you very much for pointing this out! We will correct this in the camera-ready version of the paper.

---

> > ### Comment · Reviewer_yJzi · 2025-08-01
> > **Response acknowledgement**
> >
> > I thank the authors for their response.
> > I am happy to maintain my positive evaluation of the paper.

---

### Official Review · Reviewer_Bk2z · 2025-07-03

**Clarity:** 2
**Significance:** 2
**Originality:** 2
**Rating:** 4
**Confidence:** 4

**Summary:**

This paper systematically investigates the computational complexity of counting linear regions in ReLU neural networks, a key measure of their expressive power. The authors first identify six non-equivalent definitions of linear regions used in literature, clarifying their relationships through a unified taxonomy. For shallow networks with only one hidden layer, they show that while detecting multiple regions is polynomial-time solvable, exact counting is #P-hard across all definitions. For deep networks, they significantly strengthen prior work by proving NP-hardness for any constant number of layers at least 2 and establishing strong inapproximability results. Notably, they demonstrate that even deciding whether a deep network has larger than 1 region is NP-complete. On the positive side, they prove polynomial-space upper bounds for three definitions. The work highlights fundamental limitations in analyzing neural networks through linear regions while providing rigorous complexity classifications. Key technical innovations include novel reductions from SAT/#SAT and careful handling of definitional subtleties that had caused inconsistencies in prior literature.

**Questions:**

1.Could the authors discuss whether their results imply limitations for empirical methods (e.g., sampling-based counting)? A brief discussion of practical workarounds (e.g., restricted network classes) would strengthen impact.

2.The analysis is limited to ReLU networks. Do similar hardness results hold for smooth activations (e.g., sigmoid, GeLU)?

**Ethical Concerns:**

["NO or VERY MINOR ethics concerns only"]

**Final Justification:**

After reading the rebuttal from the authors, and the discussions among other reviewers and authors, I tend to increase my score yet with unconfident evaluation. However, whether the task in this paper is suitable for NeurIPS is still an unresolved issue.

**Limitations:**

yes

**Paper Formatting Concerns:**

n.a.

**Quality:**

2

**Strengths And Weaknesses:**

Strengths:

Quality

1.The paper is technically rigorous, with well-constructed proofs and reductions (e.g., #P-hardness for shallow networks, NP-hardness for deep networks).

2.The authors carefully address inconsistencies in prior definitions of linear regions (Section 3, Table 1), which strengthens the theoretical foundation.

3.The upper-bound result (Theorem 6.1, polynomial-space counting) is a meaningful contribution, showing that some variants of the problem are not as intractable as others.

Clarity

1.The paper is well-structured, with clear distinctions between different definitions of linear regions and their implications.

2.The writing is precise, though some proofs in the appendix are dense (as expected for a theoretical paper).

Significance

1.The results have broad implications for understanding the expressive power and complexity of neural networks.

2.The hardness results (Theorems 4.2, 5.2, 5.7-5.8) suggest fundamental limitations in analyzing ReLU networks via linear regions.

Originality

1.The systematic comparison of six definitions of linear regions is novel and clarifies ambiguities in prior work.

Weaknesses:

Quality

1.Some hardness results (e.g., Theorem 5.8) rely on somewhat artificial constructions (summing multiple copies of a network), which may limit practical implications.

2.The paper does not explore whether certain restricted architectures (e.g., networks trained via gradient descent) might admit efficient counting algorithms, despite mentioning this as future work.

Clarity

1.The distinction between Definitions 4 (open connected regions) and 5 (closed connected regions) could be explained more intuitively, as their differences are subtle.

2.The proof sketches are helpful, but some steps (e.g., in Lemma 5.6) remain quite technical and may be hard to follow without careful reading.

Significance

1.While the theoretical contributions are strong, the practical implications are less clear. The hardness results suggest that exact counting is infeasible, but do not explore whether heuristic or approximate methods (beyond worst-case analysis) could be useful in practice.

2.The focus is purely on ReLU networks; extensions to other activation functions (e.g., sigmoid, GeLU) are not discussed.

Originality

1.The paper builds heavily on prior work (e.g., Wang (2022), Hanin & Rolnick (2019)), and while it refines and extends these results, the core idea of studying the complexity of linear regions is not entirely new.

2.The upper-bound result (Theorem 6.1) is somewhat incremental, as it mainly shows that some variants of the problem are in PSPACE rather than EXPTIME.

---

> ### Author Rebuttal · Authors · 2025-07-30
>
> Thank you for carefully reading and evaluating our paper, and for your constructive feedback. We hope to have addressed all questions and comments in the responses below. If you have any further questions, please don’t hesitate to post them - we’ll be happy to clarify them.
>
> >1.Some hardness results (e.g., Theorem 5.8) rely on somewhat artificial constructions (summing multiple copies of a network), which may limit practical implications.
>
> >2.The paper does not explore whether certain restricted architectures (e.g., networks trained via gradient descent) might admit efficient counting algorithms, despite mentioning this as future work.
>
> While the theoretical focus and the constructions is indeed a limitation as we acknowledged, we still think that theoretical results like Theorem 5.8 represent a valuable contribution to NeurIPS. We agree that proving results for trained networks is very interesting. This itself presents a major challenge and we are not aware of previous hardness results for such trained networks. It could very well be a topic for a new paper on its own. In our view, before trying to prove results for special cases it seems essential to first prove worst case results, which is a standard tool to understand the computational complexity of problems and an important base case. Moreover, trained neural networks often have repeating structures, so maybe our constructions are not that unnatural after all.
>
> >1.The distinction between Definitions 4 (open connected regions) and 5 (closed connected regions) could be explained more intuitively, as their differences are subtle.
>
> We acknowledge that the distinction between the two definitions is subtle and regret our explanation was not clear enough. Two main differences are that (1) closed connected regions can continue on low dimensional slices of the input space while open connected regions cannot; and (2) two open connected regions whose boundaries have a low-dimensional intersection ‘’merge’’ into a single closed connected region. These differences are illustrated in Figures 1 and 2. We will include additional details in the camera-ready version to explicitly highlight these two key differences.
>
> >2.The proof sketches are helpful, but some steps (e.g., in Lemma 5.6) remain quite technical and may be hard to follow without careful reading.
>
> We acknowledge that some of the proofs are technically involved and may require careful reading to follow in full detail. While we consider this level of rigor as natural for a theoretical paper, we are happy to incorporate any concrete suggestions that could help make the proofs or proof sketches more accessible.
>
> >1.While the theoretical contributions are strong, the practical implications are less clear. The hardness results suggest that exact counting is infeasible, but do not explore whether heuristic or approximate methods (beyond worst-case analysis) could be useful in practice.
>
> Theorems 5.7 and 5.8 show that, for Definitions 3 to 6, approximating the number of linear regions for ReLU networks with two and more hidden layers within an exponential factor is NP-hard. We acknowledge that this does not fully resolve the question, as these results are of a worst-case nature, as explicitly noted in the paper. While average-case analyses and investigation of whether networks trained via gradient descent admit heuristic or approximation lie beyond the scope of this work, we view these directions as promising fields for further research.
>
> >2.The focus is purely on ReLU networks; extensions to other activation functions (e.g., sigmoid, GeLU) are not discussed.
>
> We answer this in the dedicated question below.
>
> >1.The paper builds heavily on prior work (e.g., Wang (2022), Hanin & Rolnick (2019)), and while it refines and extends these results, the core idea of studying the complexity of linear regions is not entirely new.
>
> We fully acknowledge that our work builds upon and is motivated by prior work. In our view, the fact that linear regions have been studied extensively in earlier work strengthens the relevance of our contributions, since our results represent a substantial advancement over prior work. We emphasize that we do not merely modify existing techniques used in prior work such as those by Wang [2022] or Hanin and Rolnick [2019], but introduce novel techniques and ideas. For example, in Algorithm 2, we present a method for counting affine regions in polynomial space by enumerating all possible coefficient combinations, thereby iterating over all possible affine functions implicitly without having to store them explicitly; and in Theorem 5.8, we use a new technique that amplifies inapproximability factors by summing multiple copies of a hard network.
>
> >2.The upper-bound result (Theorem 6.1) is somewhat incremental, as it mainly shows that some variants of the problem are in PSPACE rather than EXPTIME.
>
> Previously, it was known that counting linear regions lies in EXPTIME for Definitions 1, 2 and Definition 4. While not being explicitly stated previously, devising an EXPTIME algorithm for Definition 6 is relatively straightforward. No further results were previously known, to the best of our knowledge. It remains open whether EXPTIME algorithms exist for Definitions 3 and 5. For Definition 4, PSPACE containment remains unclear. In contrast, PSPACE containment for Definitions 1 and 2 is relatively straightforward to establish, while Definition 6 requires new ideas, as outlined above. It also remains possible that PSPACE containment is the best achievable complexity bound for Definitions 1,2, and 6. Wang [2022] discusses the difficulty of proving that the problem is in NP for Definition 4, emphasizing that doing so would require fundamentally new ideas - a difficulty that seems to extend to all other definitions. Given this context, we think that our PSPACE result represents a meaningful advancement in the complexity-theoretic understanding of the problem.
>
> >1.Could the authors discuss whether their results imply limitations for empirical methods (e.g., sampling-based counting)? A brief discussion of practical workarounds (e.g., restricted network classes) would strengthen impact.
>
> Theorems 5.7 and 5.8 imply that for ReLU networks with 2 or more hidden layers, assuming NP $\neq$ P, no polynomial-time algorithm can approximate the number of linear regions - for Definitions 3 to 6 - within a factor that is exponential in the input dimension. This implies that empirical methods would require a super-polynomial number of steps (in the encoding size of the network) to provide approximation guarantees for general instances. If the input dimension is fixed, however, polynomial approximation algorithms for Definitions 3 to 6 might exist. We note that our results do not rule out the existence of approximation algorithms for Definitions 1 and 2 - although, as noted in the paper, devising such an algorithm would resolve a longstanding open problem in combinatorics.
>
> >2.The analysis is limited to ReLU networks. Do similar hardness results hold for smooth activations (e.g., sigmoid, GeLU)?
>
> Note that the function computed by a network with smooth nonlinear activation functions (e.g. sigmoid, GeLU) is itself smooth. As a result, the concept of linear regions does not apply in this setting, since such functions generally do not partition the input space into finitely many regions where the function is affine linear. However, our results do extend to other continuous piecewise linear activation functions.

---

> > ### Comment · Reviewer_Bk2z · 2025-08-08
> > **Response to Rebuttal**
> >
> > After reading the rebuttal from the authors, and the discussions among other reviewers and authors, I tend to increase my score yet with unconfident evaluation. However, whether the task in this paper is suitable for NeurIPS is still an unresolved issue.

---

### Official Review · Reviewer_gGKS · 2025-07-03

**Clarity:** 3
**Significance:** 2
**Originality:** 2
**Rating:** 5
**Confidence:** 4

**Summary:**

The work studies the problem of counting linear regions expressed by ReLU networks, from the point of view of computational complexity. The paper provides worst-case complexity results for the decision problem of determining whether a ReLU net has $K>1$ regions, and the function problem of counting linear regions.

The authors identify non-equivalent notions of affine and activation regions in the literature and relate them. Then, they provide worst-case reductions for the decision and function problems considered, for shallow and deep MLPs.

The main results can be summarized as follows:
For shallow networks, the authors show that the decision problem can be solved in polynomial time, while the counting problem cannot.
For deeper networks, time-complexity bounds are provided establishing hardness of linear region counting. Importantly, the authors present hardness results even for approximate region counting and discuss some implications for “average-case” problem instances with randomized algorithms.
Finally, improving upon prior work, the paper shows that for some notions of affine region, the function problem can be solved in polynomial space (and therefore in exponential time).

**Questions:**

I outline below a few technical questions that should be considered by the authors in order to broaden the impact of the work. I believe it is not necessary for the authors to fully address these questions in order for the paper to be accepted. However, I strongly encourage the authors to include a discussion of some of the points below in the next revision of the paper.

1. Corollary 5.5 establishes an exponential relation between input space dimensionality and the hardness of region counting. Said dependency also appears in classical results on hyperplane arrangements – most notably Zaslavsky’s Theorem –  and is connected to the well known “curse of dimensionality” in ML. I think this relation should be discussed more explicitly by the authors.

2. In the main text, it would be ideal to make explicit the dependency on $n_i$ and $L$ of Theorem 5.2, Lemma 5.3, and Theorem 5.7. Particularly, how does complexity scale as a function of $n_i$ vs $L$?

3. Hanin and Rolnick (NeurIPS’19) argue that worst-case complexity solutions in the hypothesis space of ReLU nets under random Gaussian initialization are discrete points. I wonder if the worst-case solutions you construct in your proofs fall under the same category, or whether they are stable under small perturbations of the network’s parameters.

4. I think it is of great value to draw connections between the complexity results presented in the paper and other problems in the literature that entail knowing the linear regions partition expressed by a ReLU network. A non-exhaustive list is:
  * Estimating the Lipschitz constant of ReLU networks entails minimizing a quadratic form over the underlying polytopal complex (Virmaux and Scaman, 2018).
  * Density estimation is related to the size of linear regions (Polianskii et al, 2022).
  * Computing the spectral decay of a ReLU network (Rahaman, et al. 2019)

Other minor points:

* The labeling of R1, …, R6 in the caption of Figure 1 does not match the definitions given in Section 3. Indeed, the labeling used in the caption even contradicts Theorem 3.1. This is easily fixable.

* It is common in Machine Learning literature to group hypotheses and assumptions in a dedicated paragraph, before stating technical results.  To further improve clarity and accessibility of the work, some assumptions and conjectures from computational complexity could be summarized in a dedicated section (e.g. in the appendix). It would be useful to include such a section (e.g. in the appendix) to recall common assumptions such as NP$\ne$RP.

### References

* Complexity of linear regions in deep networks. Hanin and Rolnick. ICML 2019.
* Deep relu networks have surprisingly few activation patterns. Hanin and Rolnick. NeurIPS 2019.
* Lipschitz regularity of deep neural networks: analysis and efficient estimation. Virmaux and Scaman. NeurIPS 2018.
* Voronoi density estimator for high-dimensional data: Computation, compactification and convergence. Polianskii et al. ICML 2022.
* On the spectral bias of neural networks. Rahaman et al. ICML 2019.

**Ethical Concerns:**

["NO or VERY MINOR ethics concerns only"]

**Final Justification:**

I carefully read all reviews as well as the authors responses. I believe the paper provides a solid theoretical contribution, which is generally well presented and accessible.

Before the rebuttal, my main concern with this paper was whether it is too narrow to fit a major ML conference, with more theoretical conferences being more suited (e.g. COLT). This was also expressed by one other reviewer, who suggested borderline rejection.

However, based on the two other reviews recommending acceptance, I decided to maintain my positive score.

I strongly recommend the authors to address all raised points in the next revision of the paper, to make the message clearer and more accessible, and slightly broaden the impact of the work.

**Limitations:**

Limitations of the work are properly discussed in Section 1.1. To further broaden the impact of the paper and improve contextualization of the findings in relation to the broader literature, I would have appreciated a closing discussion section describing how the technical results relate to other problems associated with linear region counting. A series of pointers is provided under “Questions” above.

**Paper Formatting Concerns:**

None.

**Quality:**

3

**Strengths And Weaknesses:**

## Strengths
1. The paper presents a nuanced discussion of worst-case computational complexity of linear region counting, under different non-equivalent notions of linear region.
2. Reductions to known hard problems (SAT and cell-counting for oriented hyperplane arrangements) are presented formally and clearly.
3. The work provides clear (counter-)examples illustrating the main technical nuances (main figures as well as appendix C). The gist of most proofs is included in the main paper.
4. The paper shows that even approximate region counting is hard for MLPs, and touches upon hardness under randomized polynomial algorithms.

## Weaknesses
1. While it is very important to establish hardness results, worst-case problem instances are limited in that they don’t provide intuition on average-case hardness for certain problem distributions that are relevant in practice. Indeed, the hypothesis space of ReLU MLPs is very large and those worst-case solutions might be rare in practice.

2. With the exception of Corollary 5.5, the dependency of the worst-case bounds on the depth and width of neural networks cannot be read directly from the main text, but only from the appendix. I believe such bounds are of independent interest beyond “region counting”, and so it would be useful to include them in the main paper.

3. Other established works in the literature (Hanin and Rolnick, ICML’19 and NeurIPS’19), argue that, under certain assumptions, “worst-case complexity” instances are isolated (discrete) points in the hypothesis space of ReLU nets (see for instance Lemma 12 in Hanin and Rolnick, NeurIPS’19). It would be useful to relate the worst-case bounds to their results (see Questions below).

4. The paper ends rather abruptly by sketching Algorithms 1 and 2. I think it is important to include a final discussion, relating the paper’s technical results with the broader literature (see Questions below). This was partially done under “Limitations”, in section 1.1.


### Justification of the given scores

**Quality:** The overall technical quality of the paper is high, with clear reductions to known hard problems. I particularly appreciate the discussion on computational hardness for randomized algorithms.

**Clarity:** The paper is overall clear. Most concepts are clearly introduced in Section 2, making the paper accessible to interested but non-specialized readers. However, the dependency on input dimension, network width, and depth is sometimes not immediately clear from the main text, and can be found only in the appendix. This could be improved.

**Significance:** The paper presents worst-case reductions for the decision and function formulations of linear region counting, for several notions of linear region as well as for shallow and deep MLPs. The work addresses some important technical inconsistencies within prior work, which are of interest beyond the narrower scope of the paper.

Most importantly, the paper discusses the hardness of exact and approximate region counting. Since region discovery appears under different guises in the ML literature (from density estimation problems to estimating Lipschitz continuity and robustness), I believe the work provides a solid base on which further theoretical analysis can be made, for specific problem distributions.

**Originality:** The work identifies a gap in the literature, whereby only limited technical results formally quantify hardness of region-counting problems (most notably Wang, 2022). The main technical novelty in terms of Mathematical proofs lies in constructing ReLU networks satisfying a given SAT condition, and then extending said construction to networks with depth $L > 1$. The proof technique was borrowed from Katz et al. (2017) as acknowledged by the authors.

Overall, I appreciate the technical contributions advanced by the work, and the only limitations I see are with the paper’s broader impact beyond the “region counting” niche.

---

> ### Author Rebuttal · Authors · 2025-07-30
>
> Thank you for carefully reading and evaluating our paper, and for your constructive feedback. We hope to have addressed all questions and comments in the responses below. If you have any further questions, please don’t hesitate to post them - we’ll be happy to clarify them.
>
> >1. While it is very important to establish hardness results, worst-case problem instances are limited in that they don’t provide intuition on average-case hardness for certain problem distributions that are relevant in practice. Indeed, the hypothesis space of ReLU MLPs is very large and those worst-case solutions might be rare in practice.
>
> We fully agree. Our paper is only a first step in this regard and we will emphasize this point further. That said, worst-case analysis is a standard tool to understand the computational complexity of problems and is an important base case.
>
> >2. With the exception of Corollary 5.5, the dependency of the worst-case bounds on the depth and width of neural networks cannot be read directly from the main text, but only from the appendix. I believe such bounds are of independent interest beyond “region counting”, and so it would be useful to include them in the main paper.
> 3. Other established works in the literature (Hanin and Rolnick, ICML’19 and NeurIPS’19), argue that, under certain assumptions, “worst-case complexity” instances are isolated (discrete) points in the hypothesis space of ReLU nets (see for instance Lemma 12 in Hanin and Rolnick, NeurIPS’19). It would be useful to relate the worst-case bounds to their results (see Questions below).
>
> We answer these comments in the dedicated questions below.
>
> >4. The paper ends rather abruptly by sketching Algorithms 1 and 2. I think it is important to include a final discussion, relating the paper’s technical results with the broader literature (see Questions below). This was partially done under “Limitations”, in section 1.1.
>
> We will add such a discussion in the revised version.
>
> >1. Corollary 5.5 establishes an exponential relation between input space dimensionality and the hardness of region counting. Said dependency also appears in classical results on hyperplane arrangements – most notably Zaslavsky’s Theorem – and is connected to the well known “curse of dimensionality” in ML. I think this relation should be discussed more explicitly by the authors.
>
> We will discuss this point in the revised version.
>
> >2. In the main text, it would be ideal to make explicit the dependency on $n_i$ and $L$ of Theorem 5.2, Lemma 5.3, and Theorem 5.7. Particularly, how does complexity scale as a function of $n_i$ vs $L$?
>
> We regret this was not clear. We will add a clarification of the dependencies in the main text. Restricting to 3SAT instances with $m$ clauses and $n\leq 3m \in \mathcal{O}(m)$ variables, then, in Lemma 5.3, we construct networks with $L=2$, width $\mathcal{O}(m)$, and input dimension $n$. The networks in Theorem 5.2 have $L$ hidden layers, width $\mathcal{O}(m + K)$, and input dimension $n$. The networks in Theorem 5.7 have $L$ hidden layers, width $\mathcal{O}(m)$, and input dimension $n$.
>
> >3. Hanin and Rolnick (NeurIPS’19) argue that worst-case complexity solutions in the hypothesis space of ReLU nets under random Gaussian initialization are discrete points. I wonder if the worst-case solutions you construct in your proofs fall under the same category, or whether they are stable under small perturbations of the network’s parameters.
>
> We think that it is important to distinguish between worst-case instances in terms of computational complexity (“hard to count”) and those in terms of the linear region count (“many regions”). The relationship between these two notions is not straightforward - a network having few regions does not necessarily imply the regions of the network are “easy to count”. In fact, for Definitions 3 to 6, our reduction in Lemma 5.6 shows that instances with relatively few regions (compared to the worst-case number) can lead to #P-hard counting problems. While Hanin and Rolnick (NeurIPS 2019) reason about instances with “many regions” (for a generalization of Definition 2), our reductions construct instances that are worst-case instances in terms of computational complexity (“hard to count”), but these instances generally do not match the worst-case number of linear regions. In fact, counting regions can be seen as a method to identify whether an instance represents a worst-case in terms of linear region count.
> It seems that the number of linear regions of the networks in our constructions is not stable under small perturbations of weights and biases for at least some of the six definitions, but making this rigorous is subject to further research.
>
> >4. I think it is of great value to draw connections between the complexity results presented in the paper and other problems in the literature that entail knowing the linear regions partition expressed by a ReLU network. A non-exhaustive list is:
> >- Estimating the Lipschitz constant of ReLU networks entails minimizing a quadratic form over the underlying polytopal complex (Virmaux and Scaman, 2018).
> >- Density estimation is related to the size of linear regions (Polianskii et al, 2022).
> >- Computing the spectral decay of a ReLU network (Rahaman, et al. 2019)
>
> We will add a discussion on how our complexity results compare to results on other problems in the literature.
>
> >The labeling of R1, …, R6 in the caption of Figure 1 does not match the definitions given in Section 3. Indeed, the labeling used in the caption even contradicts Theorem 3.1. This is easily fixable.
>
> Thank you for pointing this out! The ordering in the captions needs to be reversed. The caption for Figure 1 should read: $R_6 = R_5 = R_4 = 3, R_3 = 4, R_2 = 6$ and $R_1 = 8$. We will correct this in the camera-ready version.
>
> >It is common in Machine Learning literature to group hypotheses and assumptions in a dedicated paragraph, before stating technical results. To further improve clarity and accessibility of the work, some assumptions and conjectures from computational complexity could be summarized in a dedicated section (e.g. in the appendix). It would be useful to include such a section (e.g. in the appendix) to recall common assumptions such as NP$\neq$ RP.
>
> We will add a paragraph with hypotheses and assumptions and add explanations for complexity-theoretic assumptions.

---

> > ### Comment · Reviewer_gGKS · 2025-08-05
> >
> > I wish to thank the authors for their thorough response and for clarifying the difference between their worst-case results and those of Rolnick and Hanin (Q3). Based on the rebuttal, I maintain my positive score.
> >
> > I encourage the authors to include in the next revision of the manuscript:
> >
> > 1. Connection to Zaslavsky's theorem.
> > 2. Explicit dependency of the Theorems and Lemma presented in the main paper on $n, L, m, K$ as per the authors' rebuttal.
> > 3. Paragraph discussing "many regions" vs "hard to count" partitions.
> > 4. Relation to known problems that implicitly rely on region enumeration.

---

### Note · Authors · 2025-08-14

We thank all reviewers for their careful evaluation of our paper and their constructive feedback. While the majority recommend acceptance, one reviewer continues to question the relevance of our results to the NeurIPS community. We believe our work fits well for the following reasons:

Counting linear regions is both theoretically important and practically relevant, as demonstrated by prior works published in prominent ML venues (e.g., Hanin & Rolnick; Serra et al.; Wang) that develop and implement practical algorithms for this task - showing clear interest within the ML community. Our work substantially improves upon existing results and, for the first time, establishes the hardness of approximation, providing rigorous boundaries on computational feasibility that can guide both theoreticians and practitioners.

The NeurIPS call for papers explicitly welcomes theoretical contributions with machine learning relevance. Our results address a well-established measure of expressivity for widely used ReLU networks from a complexity-theoretic perspective, placing them clearly within NeurIPS’s scope.

For these reasons, we believe our work is a natural fit for NeurIPS and will be of clear interest to its community.

---

### Decision · Program_Chairs · 2025-09-17

**Decision:**

Accept (poster)

**Comment:**

This paper offers a comprehensive theoretical study of the computational complexity of counting linear regions in ReLU neural networks. It examines six non-equivalent definitions from the literature, clarifies their relationships, and establishes hardness results: #P-hardness for shallow networks, NP-completeness for deeper networks with a constant number of hidden layers (with corresponding inapproximability results in both cases), and PSPACE upper bounds for certain definitions.

Initial concerns, such as reliance on worst case analysis, were largely addressed in rebuttal. Some of the discussion revolved around a fundamental concern, voiced most strongly by Reviewer 9Trg, which questioned whether the problem studied constitutes a direct practical objective for the ML community. As the field of deep learning evolves, new concepts and methods (in this case established and powerful tools from theoretical computer science) are increasingly introduced to address important open questions. A recurring challenge is assessing how well such contributions fit within the broader context of deep learning theory and whether they provide practically relevant new insights. In this case, the consensus leaned toward acceptance on the grounds of the fundamental importance that linear regions play in the theoretical understanding of ReLU networks (including expressiveness and optimization), and overall the paper was praised for its systematic treatment of definitions, its rigor, and its novel hardness proofs. The authors are encouraged to incorporate the valuable comments provided by the reviewers and to further emphasize the empirical aspects underlying the study of linear regions.